# The Role of Prenatal Care in Fetal and Infant Development in Brazil: A Narrative Review

**DOI:** 10.3390/healthcare13192414

**Published:** 2025-09-24

**Authors:** Amanda Maieski da Silva, Caroline Stadler, Luiz Gustavo Gusson de Camargo, Paula Rothbarth Silva, Nathalia Marçallo Peixoto Souza, Mateus Santana Lopes, Fabiane Gomes de Moraes Rego, Juliana Sartori Bonini, Roberta Fabbri, Jéssica Brandão Reolon, Luana Mota Ferreira, Marcel Henrique Marcondes Sari

**Affiliations:** 1Department of Medicine, State University of the Midwest, Guarapuava 85040-167, PR, Brazil; amanda.maieski.med@gmail.com (A.M.d.S.); ccarolst@gmail.com (C.S.); 2Postgraduate Program in Pharmaceutical Sciences, State University of the Midwest, Guarapuava 85040-167, PR, Brazil; gustavogusson@gmail.com (L.G.G.d.C.); juliana.bonini@gmail.com (J.S.B.); roby.fabbri11@gmail.com (R.F.); 3Postgraduate Program in Pharmaceutical Sciences, Federal University of Paraná (UFPR), Curitiba 80060-240, PR, Brazil; prothbarth@gmail.com (P.R.S.); nathaliamarcallo@gmail.com (N.M.P.S.); masanlo@outlook.com.br (M.S.L.); fgmrego@gmail.com (F.G.d.M.R.); luanamota@ufpr.br (L.M.F.); 4Department of Pharmacy, State University of the Central-West (UNICENTRO), Guarapuava 85040-167, PR, Brazil

**Keywords:** child development, infant development, perinatal, primary healthcare, quality of care

## Abstract

**Background/Objectives:** In Brazil, nearly universal access to prenatal care coexists with ongoing negative fetal and infant outcomes. This review explores how the adequacy of prenatal care affects maternal, fetal, and child health, and highlights recurring gaps in service delivery. **Methods:** A narrative review of Brazilian studies published between 2018 and 2024 was conducted through the Virtual Health Library and PubMed. The initial search (July 2024) was updated in February 2025, and two reviewers independently screened and extracted data, synthesizing clinical outcomes from the findings. **Results:** A total of thirty-six studies were included in the review. Inadequate prenatal care was consistently linked to higher rates of infant and neonatal mortality, prematurity, low birth weight, congenital syphilis, and neonatal near misses. The studies indicated that counting visits alone does not adequately reflect the quality of care: when evaluated against the Prenatal and Birth Humanization Program (PHPN), most studies met only one of the eight minimum criteria. Common shortcomings included late initiation of care, incomplete diagnostic testing, fragmented follow-up, and insufficient treatment for partners regarding sexually transmitted infections. **Conclusions:** Adverse outcomes persist in Brazil not due to a lack of access, but rather due to deficiencies in the content and continuity of prenatal care. To improve perinatal outcomes, it is essential to strengthen care through standardized, multidimensional indicators and integrated strategies that combine clinical, educational, and psychosocial support.

## 1. Introduction

Prenatal care provides screening, risk stratification, counseling, and timely diagnosis and treatment that are central to health promotion and disease prevention in pregnancy [1]. Adequate prenatal care is a key determinant of maternal–fetal indicators and can reduce leading causes of maternal and neonatal mortality [2]. Beyond reducing mortality, effective prenatal care also contributes to the prevention of short- and long-term complications, including prematurity, congenital infections, and developmental disorders. Furthermore, it creates opportunities for nutritional and psychosocial support, thereby strengthening maternal well-being and influencing child health trajectories across the course of life [2].

A frequent proxy for quality in the literature is the number of visits. Yet prenatal care encompasses clinical consultations and nutritional, psychological, and educational support, and there is no full standardization of what constitutes adequate care across studies or services [3]. Current recommendations vary: the World Health Organization proposes at least eight prenatal visits during pregnancy [4], the American College of Obstetricians and Gynecologists and the American Academy of Pediatrics suggest 12 to 14 prenatal visits [5], and in Brazil a minimum of six prenatal appointments during pregnancy is recommended by the Ministry of Health [6]. Despite very high national coverage (98.7%), only 73.1% of pregnant women complete the minimum recommended number of consultations, revealing a gap between access and effective follow-up [5].

Globally, the adequacy of prenatal care remains a central concern. According to the World Health Organization, inadequate prenatal follow-up contributes substantially to preventable maternal deaths, stillbirths, and neonatal complications, with prematurity and low birth weight ranking among the leading causes of neonatal mortality worldwide [7]. In high-income countries, emphasis has shifted from visit counts toward comprehensive, multidimensional indicators of care, including timely initiation, diagnostic completeness, and psychosocial support [8]. However, in low- and middle-income countries, structural barriers such as limited health system capacity, socioeconomic inequality, and fragmented care pathways persist, reinforcing the global challenge of translating access into effective outcomes. Within this broader landscape, Brazil reflects a paradox that is epidemiologically relevant beyond its borders: near-universal coverage coexists with persistent adverse perinatal outcomes, making the country an important case for examining how quality gaps undermine the potential benefits of prenatal care [9].

Brazil has made significant progress with the Prenatal and Birth Humanization Program (PHPN), which establishes minimum standards for comprehensive care. This includes initiating prenatal care in the first trimester, conducting at least six medical consultations, and ensuring the completion of essential laboratory tests and preventive measures. The PHPN serves as an appropriate reference framework for evaluating the quality of routine practices in this area [9]. Despite advancements in maternal healthcare, several limitations continue to compromise the quality and continuity of care. These include delayed initiation of prenatal services, inadequate execution of essential assessments, disjointed follow-up throughout the pregnancy-postpartum continuum, and ongoing regional and socioeconomic disparities that restrict equitable access to care [10]. The result is a scenario in which visit counts alone do not capture quality, and process indicators become critical for interpreting outcomes.

Evidence consistently links inadequate prenatal care to immediate adverse outcomes such as prematurity and low birth weight [9] and to longer-term consequences for infant development, given that early-life conditions shape health and well-being across the life course [11]. This paradox, high coverage coexisting with unfavorable outcomes, raises a question that is both epidemiologically relevant and actionable for policy. It reflects the persistent gap between access and quality, in which the number of visits alone does not ensure the delivery of essential diagnostic, preventive, and counseling components of prenatal care. Addressing this discrepancy requires multidimensional indicators that capture not only the frequency but also the content and continuity of care, thereby guiding interventions that can effectively improve maternal and child health outcomes.

In this context, and recognizing the persistence of unfavorable fetal, neonatal, and child health outcomes within the Brazilian healthcare system despite high prenatal care coverage, this review aims to explore the nuances of this reality. This study is justified by its contribution to providing a deeper perspective on the importance of prenatal care, with an emphasis on fetal and child health, as well as by identifying critical areas that still require improvement in the quality of care delivery. Although the importance of prenatal care for maternal and neonatal outcomes is widely recognized globally, the specific characteristics of healthcare systems, socioeconomic factors, and implementation challenges in middle-income countries such as Brazil often result in gaps in the quality of care that require in-depth investigation. This scenario is further complicated by the persistence of adverse neonatal outcomes, despite theoretically high prenatal care coverage.

Therefore, this narrative review aims to synthesize Brazilian scientific evidence to provide a contextualized understanding of how the quality of prenatal care correlates with specific outcomes in fetal and child development. Our objective is not to reaffirm the well-established association between prenatal care and infant health, but rather to explore where and how shortcomings in the provision of prenatal care in Brazil contribute to the persistence of these public health challenges, offering a critical overview to inform the improvement of local policies and practices. By synthesizing Brazilian evidence within this broader global framework, our review contributes to understanding how systemic gaps in prenatal care translate into adverse outcomes, offering insights not only for national policies but also for other middle-income settings facing similar challenges.

## 2. Materials and Methods

A comprehensive literature review aimed to determine the current knowledge on a specific topic by identifying, analyzing, and synthesizing results from independent studies employing various methodologies on the same subject [12]. This approach provides a comprehensive overview of complex concepts, theories, or relevant health issues [13]. In this regard, the search was specifically tailored to capture specific nuances and critical insights from the Brazilian context.

### 2.1. Design and Scope

We conducted a narrative review to identify, analyze, and synthesize Brazilian evidence on how the adequacy of prenatal care relates to fetal and infant outcomes. This design allows aggregation of heterogeneous observational literature and a contextualized appraisal of health-system specificities that are often diluted in global syntheses. The review explicitly prioritizes Brazilian evidence to capture nuances of the SUS and the national socio-demographic landscape. Searches were run in July 2024 and updated in February 2025 in the Virtual Health Library and PubMed. We used the descriptors “prenatal care”, “antenatal care”, “child development”, and “infant development”, combined with Boolean operators (OR/AND).

### 2.2. Guiding Question and Eligibility Criteria

This review was guided by the question “How does the inadequacy of prenatal care affect fetal and child development in Brazil?”, with a priori eligibility focused on Brazilian evidence to capture health-system specificities that are frequently diluted in global syntheses. Accordingly, we included observational, qualitative, or evaluation studies published in Portuguese between 2018 and 2024 that examined neonatal and/or infant outcomes in relation to prenatal care adequacy; when an explicit adequacy construct was not available, at minimum the frequency of prenatal consultations had to be reported as a proxy. We excluded studies centered exclusively on maternal outcomes, studies coupling prenatal care with additional therapeutic interventions beyond routine care, and records lacking a clear definition or proxy for “adequate prenatal care”. Reviews, conference abstracts, and dissertations were not eligible, nor were articles published before 2018 or in languages other than Portuguese. This language-and period-delimitation was intentional to ensure a contextualized assessment of SUS-related practices and constraints that could otherwise be overlooked.

### 2.3. Study Selection and Data Extraction

Two reviewers independently and blindly screened titles/abstracts and then full texts using Rayyan. Disagreements were resolved by consensus. We documented the selection flow in a diagram (Figure 1). For each eligible study, we extracted: author and year, study design, sample size, prenatal care adequacy metric (e.g., timing and number of visits, testing, counseling), and neonatal/infant outcomes. Extraction was performed into a structured table (Table 1), followed by a critical narrative synthesis organized by the primary clinical outcome per study.

### 2.4. Operationalization of Prenatal Care Adequacy

Given the heterogeneity of metrics across studies, we harmonized prenatal care adequacy using the Brazilian Prenatal and Birth Humanization Program (PHPN) as a reference framework. When possible, each study was mapped to the number of PHPN criteria fulfilled; when only visit counts were available, frequency served as a proxy of adequacy, with its limitations explicitly noted in the synthesis. This operationalization underpins the comparative analysis presented later (see Figure 2C).

### 2.5. Outcomes and Synthesis Approach

Outcomes of interest included those reported by eligible literature and subsequently grouped for analysis: congenital syphilis; prematurity; low Apgar score at 5 min (<7); neonatal near miss; neonatal and infant mortality; low birth weight and later excessive weight gain; congenital anomalies; miscarriage; hospitalization; and risk indicators for hearing impairment (RIHI). Results were synthesized narratively by outcome domain, emphasizing comparability of adequacy definitions and highlighting recurrent process gaps (late initiation, incomplete testing, fragmented follow-up, limited partner management in STI care).

### 2.6. Quality Considerations

Because this is a narrative review, no formal risk-of-bias tool was systematically applied. However, data was reported when individual studies adjusted for confounders, and the findings interpreted considering heterogeneity in design, samples, and adequacy metrics. The synthesis explicitly distinguishes coverage/visit counts from content/continuity of care to avoid over-reliance on frequency as a quality surrogate.

## 3. Results

The search identified 45,258 records. After an initial relevance filter, 393 records proceeded to screening; 40 remained eligible at title/abstract level and were assessed in full. Of these, 23 met all criteria from the database search, and handsearching contributed 13 additional studies, yielding 36 studies. These results are illustrated in Figure 1.

We analyzed quantitative publications linking prenatal care quality criteria to neonatal and infant outcomes (Table 1). Figure 2A depicts the temporal distribution of included studies, and Figure 2B groups them by outcome: congenital syphilis; low birth weight and later excessive weight gain; infant mortality; low Apgar score and neonatal asphyxia; neonatal near miss; prematurity; congenital anomalies; miscarriage; hospitalization and mortality; risk indicators for hearing impairment (RIHI); and multi-outcome evaluations. Finally, using the Prenatal and Birth Humanization Program (PHPN) as a reference, studies were mapped by the number of minimum care criteria fulfilled (Figure 2C). Most fulfilled only one of the eight criteria (21/36; 58%), and the maximum observed was four criteria (5/36; 14%), underscoring that visit counts alone capture coverage but not the content and continuity of care.

Eleven studies evaluated congenital syphilis in relation to prenatal processes, including attendance and number of visits, timing of maternal and infant diagnosis, and maternal and partner treatment [15,19,21,23,28,30,36,37,42,44,47]. Collectively, these analyses link shortcomings in screening and case management to adverse neonatal endpoints, highlighting how late or incomplete testing, inadequate maternal therapy, and insufficient partner management undermine vertical transmission control. The body of evidence also characterizes associated factors such as prematurity, the demographic profile of affected pregnant women, spatial distribution of cases, and operational barriers faced by health professionals and patients to interrupt transmission. Two studies addressed weight deviation (low birth weight and later excessive weight gain) using the number of prenatal consultations as a quality parameter [14]. Insufficient prenatal attendance emerged as a relevant contributor to increased preschool weight gain, reinforcing the role of early counseling on breastfeeding and healthy complementary feeding. Regarding low birth weight, the studies emphasize that timely access to information and continuous nutritional follow-up from the start of pregnancy are central to prevention [17,31,35].

Four studies examined neonatal and infant mortality in relation to prenatal care and consistently found higher mortality where prenatal follow-up was insufficient, with effects modulated by care setting quality at birth, prematurity, low birth weight, maternal history of previous infant death, and socioeconomic conditions [29,32,39,43]. Another four studies evaluated low Apgar scores in connection with the timing and number of prenatal visits, alongside gestational age and birth weight, assessing outcomes at the first and fifth minutes after birth [18,24,38,46]. Across this evidence, prematurity, low birth weight, and congenital anomalies emerged as the strongest risk factors for an Apgar score ≤ 7 at five minutes. Three studies reported neonatal near miss outcomes and associated them with prenatal care adequacy as defined within each study, typically combining visit frequency with documentation of routine examinations and procedures [16,22,45]. Collectively, these findings suggest that content and continuity of care, beyond visit counts, are critical to reducing the risk of severe, near-fatal neonatal events. Two studies identified prematurity as an outcome linked to the number and initiation timing of prenatal consultations, as well as to the presence of congenital syphilis [26,34]. In addition, two publications related congenital anomalies to the number of prenatal visits, showing that lower gestational age, lower Apgar scores, and lower birth weight were associated with higher anomaly rates in the studied populations [33,41].

One descriptive study examined spontaneous abortion (miscarriage) and linked higher abortion rates to fewer prenatal visits, underscoring the importance of early enrollment and continuous follow-up [18]. Neonatal asphyxia, defined as an Apgar score < 6 at five minutes, was likewise analyzed in relation to the number of prenatal visits, reinforcing the role of adequate prenatal monitoring for perinatal safety [38]. In addition, one study evaluated risk indicators for hearing impairment (RIHI) in term and preterm infants and associated higher risk with fewer prenatal consultations, particularly among preterm newborns [25]. Finally, three studies reported multiple outcomes—including prematurity, low birth weight, fetal death, Apgar at the 1st and 5th minutes, neonatal near miss, respiratory distress, and oligohydramnios—and related these endpoints to prenatal care adequacy [26,34,40].

The next section provides an integrated overview of these findings within the Brazilian healthcare context, highlighting salient interrelations among outcomes and discussing their implications for process quality in prenatal care. This synthesis underscores that adverse outcomes rarely occur in isolation, but rather cluster through shared causal pathways, such as inadequate screening, late initiation of care, and incomplete follow-up. By examining these patterns collectively, the discussion aims to provide insights not only into clinical consequences but also into the systemic challenges that must be addressed to improve maternal and child health in Brazil.

## 4. Discussion

### 4.1. Miscarriage

According to the Ministry of Health’s Technical Standard on Humanized Care for Abortion, abortion is defined as the interruption of pregnancy up to the 20th–22nd week with a product of conception weighing less than 500 g [49]. In global terms, an estimated 121 million unintended pregnancies occurred annually between 2015 and 2019; not all resulted in abortion, and the proportion of unintended pregnancies increased from 51% to 61% between 1990 and 2019 [50]. Moreover, among the 56 million women who underwent abortion annually from 2010 to 2015, 25 million procedures were classified as less safe, indicating that abortion-related complications remain a major driver of maternal mortality [51]. Notwithstanding this burden, evidence specifically linking prenatal care adequacy to spontaneous abortion (miscarriage) remains limited in the Brazilian literature.

In the study by Carvalho and collaborators, which analyzed 147 women with a diagnosis of miscarriage, most participants had no prenatal consultations [20]. Absence of prenatal care was associated with increased gestational complications, underscoring the importance of early enrollment to mitigate risks to maternal and fetal health. These findings align with international evidence showing that inadequate or absent prenatal care is associated with worse outcomes, including recurrent miscarriage and maternal mortality [51]. While causal inference is limited by observational design and potential confounding by socioeconomic and access-related factors, the observed pattern is consistent with the broader literature on the protective role of timely, continuous, and content-adequate prenatal care.

From a public health perspective, strategies that promote early booking and sustained engagement, such as proactive outreach and home visits by family health teams, along with targeted health education, may enhance adherence among vulnerable populations [20]. Humanized care and psychosocial support are likewise relevant to reducing complications and improving patient experience. Finally, implementing quality indicators for prenatal care and ensuring that each consultation delivers the PHPN-recommended content (screening, testing, counseling, and follow-up) could support systematic monitoring and improvement of services, moving beyond visit counts toward measurable adequacy of care [20].

### 4.2. Congenital Syphilis

The global prevalence of gestational syphilis in 2016 was 0.69%, and the global rate of congenital syphilis reached 473 cases per 100,000 live births, totaling 661,000 cases [52]. The World Health Organization has implemented strategies to expand diagnosis and treatment in pregnancy, with the goal of reducing congenital syphilis to fewer than 50 cases per 1000 live births in at least 80% of countries by 2030 [53]. In Brazil, the 2023 Syphilis Epidemiological Bulletin reported 86,111 notifications of syphilis in pregnant women, 25,002 cases of congenital syphilis, and 196 deaths due to congenital syphilis [54]. In alignment with this burden, the Ministry of Health recommends prevention through prenatal care that ensures appropriate follow-up for diagnosed pregnant women and adequate treatment of sexual partners.

Across most studies on congenital syphilis, a large share of women attended at least one prenatal visit, as reported by some studies included in this review [19,21,23,42]. However, visiting frequency alone does not guarantee quality. Effective prenatal care must identify risks, provide guidance, ensure timely treatment, and make appropriate referrals throughout pregnancy to reduce maternal–infant morbidity and mortality [10].

Screening coverage varies widely among the selected studies. Araújo and collaborators reported that 71.1% of women underwent at least one syphilis test [15]; the study of Caldeira and colleagues found treponemal testing in 35.3% of women [19]; and other study documented non-treponemal testing in 83.33% [44]. In Bahia, Soares and coworkers showed that in 2017 only 1 of 123 municipalities performing rapid tests met the national goal of having 95% of pregnant women tested twice during prenatal care [47]. According to the Brazilian Ministry of Health, screening should combine treponemal and non-treponemal tests, preferably starting with a rapid treponemal test, offered to all pregnant women at least in the first and third trimesters or whenever exposure is suspected. The reviewed studies nevertheless revealed gaps in test uptake and information, indicating shortcomings that compromise follow-up quality.

The timing of diagnosis was frequently suboptimal. Most diagnoses occurred in the second trimester [19]. In some studies, an average of 62.1% of cases were identified during prenatal care [23,28,30,44]. Favero and colleagues reported that 77.67% of children with congenital syphilis were born to mothers diagnosed during pregnancy [21], whereas other study found that only 41.3% were diagnosed during prenatal care [37]. High proportions of diagnoses made at delivery persist across studies, favoring vertical transmission and reinforcing the need for earlier identification.

The mean proportion of inadequately treated mothers was 65.47% as reported by numerous studies included in this review [21,23,28,30,36,37,42,44]. Although benzathine penicillin is available in primary care to prevent vertical transmission, one study indicated that many Brazilian states have a high prevalence of primary care services with inadequate conditions for diagnosis and treatment, hampering early identification of Treponema pallidum in pregnant women [54,55]. These structural constraints directly threaten prevention efforts.

Partner treatment remains a critical bottleneck. On average, only 32.56% of partners received adequate therapy [19,21,28,37,42,44]. Communication barriers, including fear of the partner’s reaction, may contribute to under-treatment, and even when informed, few partners complete therapy. This scenario necessitates targeted interventions and an analysis of women’s difficulties in disclosing the diagnosis to avoid treatment failures [56]. Reinfection is a tangible risk, which was reported by Caldeira and colleagues (reinfection rate of 20.2%) [19], underscoring the need to systematically include partners in prenatal pathways.

Adverse neonatal outcomes were frequent in the presence of maternal infection and gaps in care. Around 75% of newborns were premature among mothers who attended prenatal care, and 37.5% of these women had not undergone any diagnostic testing for syphilis [15]. Furthermore, the studies reported fetal deaths and two miscarriages [19], deaths due to congenital syphilis [21], miscarriages and stillbirths among congenital syphilis cases [36,37,44]. Extending beyond syphilis, Silva and collaborators demonstrated significant associations between several vertically transmissible infections (HIV, hepatitis B and C, Zika virus, toxoplasmosis, and syphilis) and unfavorable fetal or neonatal outcomes, including respiratory distress, hypoxia, sepsis, congenital malformations, amniotic fluid abnormalities, and altered infant size; notably, 88.5% of complicated cases had fewer than four prenatal visits [31]. Together, these findings reinforce that preventing congenital syphilis and its sequelae depends on service adequacy, including timely screening, definitive diagnosis, complete maternal therapy, systematic partner treatment, and documented follow-up, rather than attendance alone.

### 4.3. Low Birth Weight and Excessive Weight Gain in Childhood

Low birth weight, defined by the WHO as <2500 g, is a major global public health concern and is associated with multiple short- and long-term outcomes [1,2]. More than 80% of the 2.5 million newborn deaths each year occur among low-birth-weight infants [57]. Birth weight is therefore a sentinel marker of intrauterine conditions and the most influential individual factor for neonatal survival and early health [58]. In a Brazilian cohort of over 17 million births (2011–2018), 9.6% were low birth weight, and these newborns had a 25-fold higher risk of neonatal mortality compared with those of normal weight [59].

In Brazil, the Ministry of Health recommends at least six prenatal consultations to safeguard maternal and neonatal well-being [1,2]. Accordingly, a study reported that improvements in prenatal coverage reduced the risk of low birth weight across all regions [60]. In a sample of 751 adolescents, fewer than six consultations was associated with a higher likelihood of low birth weight [17]. Regarding nutritional assistance during pregnancy, regular intervention was linked to a lower occurrence of low birth weight and to reduced risks of gestational diabetes, preeclampsia, and prematurity [61]. Silva and collaborators found that the number of prenatal visits and maternal infections were associated with neonatal outcomes, including low birth weight in 39% of cases, with 10.8% of infants small for gestational age [31]. Moreover, a lower number of consultations increased the odds of low birth weight by 2.35 times, underscoring the importance of more frequent and timely visits to enable effective interventions [35].

Childhood obesity is another pressing public health issue, characterized by excess adiposity and weight gain driven by environmental, psychosocial, economic, biological, and behavioral factors [6,9]. Cardiometabolic risk in children is assessed through anthropometric indicators [62]. In an analysis of conditional weight gain among 326 children, those born to mothers with fewer than six prenatal consultations exhibited higher-than-expected weight gain for age, suggesting missed opportunities for counseling on breastfeeding and healthy complementary feeding when visit frequency falls below recommendations [14]. As noted by Tourinho and colleagues, routine weight monitoring during prenatal care is essential to detect nutritional deviations and implement corrective plans; inadequate dietary support in pregnancy, whether insufficient or excessive, can alter intrauterine development and culminate in inappropriate birth weight [63].

Together, these findings indicate that translating coverage into better outcomes requires not only sufficient visit frequency but also the consistent delivery of nutrition-focused content, timely risk identification, and longitudinal follow-up throughout pregnancy.

### 4.4. Inadequate Child Development

Childhood is a sensitive period during which children acquire the competencies that scaffold subsequent developmental stages [1,4]. When the early years unfold under favorable conditions, they enable children to approach their full developmental potential. Development is commonly described across four core domains: motor, language, cognitive, and social. Regular follow-up by healthcare professionals allows early identification of alterations or delays in any of these domains and timely referral for intervention [64]. The high proportion of children with suboptimal development underscores the need to invest in child development promotion and in actions that strengthen the adult-child bond, whose quality influences developmental trajectories beginning in the prenatal period [48].

### 4.5. Risk Indicators for Hearing Impairment

Permanent hearing loss in childhood is a public health concern [65]. In a meta-analysis with a pooled data from 1,799,863 infants across multiple countries, the prevalence was 1.1 per 1000 children screened and was 6.9 times higher among infants admitted to neonatal intensive care units (NICU) [66]. In Brazil, the reported prevalence is 1 per 31,000 live births [67]. Notably, approximately 60% of pediatric hearing loss has preventable causes amenable to public health measures, including neonatal hearing screening [67].

Within this context, Nascimento and colleagues evaluated 87 infants with risk indicators for hearing impairment (RIHI), most of whom were preterm (66.7%) [25]. Lower gestational age was associated with increased auditory risk. The study also observed that mothers of preterm infants attended fewer prenatal consultations than the mothers of term infants, adding an access and adequacy dimension to risk stratification. Furthermore, inadequate prenatal care was linked to missed investigation and management of infections associated with RIHI—such as toxoplasmosis, syphilis, varicella, and HIV—and to insufficient counseling on behaviors that can harm or protect neonatal auditory health [25].

Taken together, these findings suggest that both adherence to and adequacy of prenatal care influence hearing-related outcomes in early life. The authors underscore the need for greater investment to ensure timely access and content-rich prenatal care capable of preventing conditions that threaten hearing and language development [25].

### 4.6. Neonatal near Miss

A neonatal near miss is defined as a serious event that almost resulted in the death of the newborn, occurring within the first 7 days of life [68], or up to 28 days [69]. This concept is valuable for identifying risk factors associated with neonatal mortality and contributes to assessing and improving the quality of care.

A neonatal near miss is a severe, life-threatening condition that almost results in neonatal death within the first seven days of life [68], or within 28 days [69]. The construct is valuable for identifying risk factors for neonatal mortality and for assessing and improving the quality of care.

Standardization remains challenging because health-system capabilities and evaluation structures differ widely between high-income and middle- or low-income settings [70]. To address this, the WHO conducted two multicenter cross-sectional studies—the Global Survey on Maternal and Perinatal Health (2004–2008) [71] and the Multicountry Survey on Maternal and Newborn Health (2010–2011) [72]—whose analyses underpinned the pragmatic criteria proposed by Pileggi-Castro [66], including Apgar score at five minutes < 7, birth weight < 1750 g, and gestational age < 33 weeks.

Despite these advances, conventional NNM evaluation has not been uniformly adopted. The three studies included in this review applied variable criteria, leading to differences in sensitivity, specificity, and the number of newborns classified as NNM [16,22,45]. A study explicitly notes that the usefulness of NNM as a severity gradient depends on the conceptual and operational scope adopted [22].

Across the three studies, commonly used conceptual criteria comprised Apgar score < 7 at five minutes, low birth weight, and prematurity. Two studies used <1750 g as the low-birth-weight cutoff [16,22], whereas Pereira and collaborators used <1500 g [45]; gestational age cutoffs were <33 weeks for Assis et al. and França et al., and <32 weeks for Pereira et al. Assis et al. incorporated additional management-based indicators of severe neonatal morbidity (e.g., specific interventions), yielding the most comprehensive set of criteria among the three [16,22,45].

Regarding prenatal care, a study reported a significant difference in the number of prenatal consultations among cases with reported NNM in a Pernambuco hospital [22]. Two studies identified similar inadequate prenatal care as a risk factor for NNM, underscoring the role of prenatal follow-up in the early detection and treatment of maternal conditions that affect fetal well-being, as well as conditions present in the fetus [16,45]. Importantly, Pereira and colleagues cautioned that visit counts alone do not guarantee quality, emphasizing early initiation by the 12th gestational week, qualified professionals, adequate resources, completion of recommended tests, and timely treatment when indicated [45]. As discussed throughout this review, the absence of standardized adequacy metrics complicates cross-study comparisons.

Collectively, the evidence indicates that adequate access to high-quality prenatal and childbirth care can prevent unfavorable neonatal outcomes, including NNM, primarily by ensuring timely risk identification, appropriate diagnostic work-ups, definitive treatment, and documented follow-up pathways [16,45]. This positions the quality of prenatal care—not merely coverage or frequency of visits, as a key determinant to reduce neonatal mortality and near-miss events.

### 4.7. Infant and Neonatal Death and Hospitalization

Infant mortality is a sentinel indicator of population health and living conditions, with higher rates typically reflecting socioeconomic deprivation and limited access to quality care [73]. Globally, the neonatal mortality rate (0–27 days) fell 44% from 2000 to 2022, while in Brazil the annual decline averaged 3.6% from 1990 to 2022 [74]. In the United States, the perinatal mortality rate was 5.54 per 1000 in 2021 [75]. In Brazil, 21,582 neonatal deaths were recorded in 2023, a marked reduction from approximately 101,000 in 1990, yet many remain preventable through essential, standardized, and often low-cost interventions such as high-quality prenatal care.

Across Brazilian studies, inadequate prenatal care consistently emerged as a risk factor for neonatal mortality. Some studies reported higher mortality with fewer visits, with odds ratios of 4.77 for 4–6 visits and 13.1 for ≤3 visits [29,32,39,43]; the risk remained elevated in the post-neonatal period (28–364 days) with an OR of 4.16 [29,32]. Considering deaths under one year, Maia and colleagues identified a low number of prenatal visits as the factor most strongly associated with infant mortality across all Brazilian regions (OR = 1.8) [9]. Low adherence to prenatal also correlated with newborn hospitalization [43].

Mechanistically, prenatal care protects newborns by enabling early identification and management of maternal and fetal conditions that precipitate adverse outcomes [43]. By addressing obstetric complications and ensuring timely interventions, prenatal care reduces prematurity and low birth weight, thereby decreasing the need for postnatal hospitalization and lowering infant mortality [9,43].

However, quantity alone does not equal quality. Adequacy must consider not only the recommended number of visits but also laboratory testing, timely management of complications, and continuity of care [29,32]. Even in settings with high primary care coverage, gaps persist; in Florianópolis, despite 100% coverage, 27% of 15,879 mothers had fewer than seven visits, indicating missed opportunities for content-rich care [39].

In summary, the evidence underscores prenatal care as a determinant of maternal–infant health. To translate coverage into survival gains, managers should prioritize resources that ensure quality and continuity of prenatal and childbirth care, with emphasis on early initiation, complete testing, and effective management of conditions that drive infant mortality.

### 4.8. Prematurity

Preterm birth is defined as delivery before 37 completed weeks of gestation and is classified by gestational age as extreme (22 to <28 weeks), severe (28 to <32 weeks), and moderate to late (32 to <37 weeks) [76], with the latter being the most frequent category [77]. Globally, an estimated 13.4 million babies were born preterm in 2020 [76]. In Brazil, between 2012 and 2022, 11.26% of births were preterm, placing the country tenth worldwide in absolute numbers of preterm births [77,78].

Prematurity is a WHO-defined urgent challenge due to its intrinsic link to neonatal morbidity and mortality [79]; worldwide, its complications account for approximately 29% of neonatal deaths [77]. In Brazil, prematurity remains one of the principal risk factors for neonatal and post-neonatal mortality [9,29,32,43]. Beyond survival, generalized immaturity can affect any organ system, leading to complications across development and imposing a substantial economic burden due to the need for higher-complexity care [80].

Inadequate prenatal care influences preterm birth through multiple pathways, intersecting with socioeconomic and family vulnerability, reproductive conditions, and pregnancy non-acceptance [26]. Consistently, fewer prenatal visits and late initiation have been associated with higher odds of late preterm birth (qualitative direction), and among nine women without any prenatal care in Vanin et al. (2020), seven delivered preterm [35]. In Leal and colleagues, inadequate prenatal care was linked to both spontaneous and provider-initiated preterm birth [40]. Prenatal care also reflects social, economic, and psychological conditions [26]. Alcohol consumption during pregnancy, more frequent among women with late initiation and an inadequate number of visits, exerted direct effects and indirect effects via pregnancy non-acceptance and insufficient prenatal care, increasing the risk of late prematurity by about threefold [81].

Importantly, the observed association between fewer visits and prematurity is partly shaped by gestational truncation, given that preterm pregnancies have less time to accrue visits [34]. Even so, incomplete prenatal care, which considers not only the number of visits but also timely initiation, completion of recommended tests and procedures, and essential counseling, emerged as a mediating factor for preterm birth [26]. Altogether, the evidence indicates that inadequate prenatal care acts as a significant risk factor for preterm birth through cumulative relationships with socioeconomic, psychological, and behavioral vulnerabilities [33,36]. Strengthening the quality and comprehensiveness of prenatal care is therefore expected to reduce prematurity and, consequently, infant morbidity, mortality, and the healthcare system’s economic burden.

### 4.9. Low Apgar Score

The Apgar score is a standardized newborn assessment that guides obstetric and neonatal decision making by evaluating heart rate, respiratory effort, reflex irritability, muscle tone, and color [82]. Each domain is scored from 0 to 2, totaling up to 10 points [83]. The total score classifies neonatal status as low (Apgar 0–3), intermediate (Apgar 4–6), or standard (Apgar 7–10) [84]. A low score at one minute is commonly associated with transient depression, whereas an unsatisfactory score at five and ten minutes generally indicates clinically significant compromise and an inadequate response to resuscitation [84]. Although low five-minute Apgar scores are better population-level predictors of neonatal and infant mortality in the first year of life [27], the American College of Obstetricians and Gynecologists and the American Academy of Pediatrics emphasize that the Apgar score alone does not reliably predict future neurologic outcomes at the individual level [83].

In a Swedish cohort, 5.3% of preterm newborns had Apgar 5 min < 7 [84]. In Brazil, the prevalence of five-minute Apgar < 7 ranged from 0.6% between 2001 and 2012 [46] to 0.9% in 2018–2019 [24]. Interpretation warrants caution due to heterogeneity in sample size and population structure. Magalhães and collaborators analyzed 5,680,092 live births, Cnattingius and coworkers evaluated 113,000 preterm infants, and Santos and collaborators examined 9135 newborns [24,46,85]. Differences in statistical power and precision can limit generalizability in small samples, while very large samples may detect statistically significant differences in limited clinical relevance [86,87].

Evidence from Brazil links prenatal care to Apgar outcomes. Two studies identified an insufficient number of prenatal consultations as a risk factor for low five-minute Apgar [18,24,38]. Adequate prenatal care is associated with reductions in maternal and infant mortality through timely detection and management of conditions that precipitate maternal-fetal complications [18,38]. Prenatal care also offers a platform for preventive interventions and counseling on healthy behaviors, birth preparation, and the postnatal period [24]. Accurate determination of gestational age, preferably by first-trimester ultrasound, is another key component; it supports appropriate referral and closer monitoring and is associated with lower risk of adverse neonatal outcomes [24,46].

Two discrepancies emerged among including studies. First, the predominance of inadequate prenatal care varies. Some authors reported that most women met the recommended number of visits (65.8%, 52.99%, and 76%, respectively), which did not diminish the observed association between prenatal follow-up and Apgar 5 < 7 [18,27,38]. In contrast, one third of the Santos and collaborators cohort had inadequate consultations [46]. Second, Magalhães and colleagues did not identify adolescence as a risk factor for low Apgar, whereas another two studies found positive associations between adolescence, low Apgar, and perinatal asphyxia, likely reflecting socioeconomic vulnerabilities that hinder consistent prenatal attendance and continuity of care [18,24,38].

Overall, the studies converge on a central message. The persistence of low Apgar scores despite high prenatal coverage is more closely related to care quality than to visit counts alone. Timely initiation, adequate number of consultations, completion of basic procedures and tests, qualified teams, and effective management of complications are critical parameters to improve neonatal vitality and reduce maternal and infant mortality [18,24,38].

### 4.10. Congenital Anomalies

Congenital anomalies comprise a broad spectrum of structural or functional alterations of prenatal origin that may be identified before birth, at birth, or later in life [88]. Globally, an estimated 6% of babies are born with a congenital disorder, contributing to approximately 295,000 neonatal deaths each year due to these anomalies or their complications [89]. In Brazil, between 2010 and 2021, congenital malformations accounted for 0.83% of live births, about 83 per 10,000 live births in the period assessed [90].

In Rio Grande do Sul, Trevilato and colleagues reported prevalences of 0.92–0.93% when evaluating all live births with anomalies from 2012 to 2015 (n = 5250), while Luz and coworkers observed similar magnitudes in a larger population of 1,386,803 live births from 2005 to 2014 [33,41]. Both studies identified associations between the number of prenatal consultations and the occurrence of congenital anomalies, with no prenatal care increasing the likelihood of congenital disorders by 97% [33]. Luz et al. further showed that seven or more prenatal visits were associated with the lowest anomaly rates, suggesting a protective effect [41]. Supporting these findings, other study documented a 2.6-fold higher prevalence ratio of microcephaly at birth among mothers without prenatal care when compared with those who attended six or more consultations [91].

Mechanistically, adequate prenatal care enables primary prevention through actions recommended in routine practice, including guidance on iodine and folic acid supplementation folic acid as prophylaxis for neural tube defects and adherence to appropriate vaccination schedules to prevent gestational infections [33,92]. It also provides a platform for behavioral counseling on nutrition, weight control, medication use, alcohol, drugs, and other teratogens, and for the timely monitoring and treatment of maternal conditions such as hypertension, diabetes mellitus, epilepsy, hypothyroidism, and HIV/AIDS [93]. Consistent attendance further empowers women during pregnancy, fostering informed decision making, earlier anomaly detection, and timely referral when needed [41].

Overall, the evidence indicates that the frequency of prenatal visits is associated with anomaly prevalence, but the protective effect arises from the content and continuity of care delivered during those visits. Early initiation, sufficient consultations, completion of recommended tests and preventive measures, and effective counseling converge to reduce the burden of congenital disorders and improve perinatal outcomes [33,41].

### 4.11. Expert Opinion and Perspectives

Across the included studies, beyond the influence of prenatal care quality, we observed direct interrelationships among outcomes, whereby several endpoints acted as risk factors for one another (Figure 3). These patterns suggest shared causal pathways and potential mediators, reinforcing that downstream neonatal outcomes often cluster rather than occur in isolation.

Centrally, prematurity and low birth weight emerged as pivotal nodes: both were associated with neonatal near miss, infant mortality, neonatal hospitalization, five-minute Apgar < 7, congenital syphilis, risk indicators for hearing impairment (RIHI), and language-acquisition outcomes, and they also influenced each other. Low one-minute Apgar (<7) and low five-minute Apgar (<7) were associated with infant mortality; additionally, five-minute Apgar < 7 correlated with neonatal hospitalizations. Congenital malformations were linked to neonatal and post-neonatal mortality, hospitalizations, and five-minute Apgar < 7; a notable adverse condition associated with malformations was a history of prior miscarriages or fetal losses. Maternal syphilis was associated with prematurity, neonatal near miss, fetal and infant mortality, miscarriage, and lower scores on language-acquisition assessments (e.g., DENVER Phase 1 at 3–4 months; SEAL-2 at 7–13 months). Taken together, the network depicted in Figure 3 underscores that improving prenatal care adequacy can reverberate across multiple interconnected endpoints rather than shifting a single outcome in isolation.

A comparative analysis of neonatal outcomes in relation to the adequacy of prenatal care is vital for understanding the actual impact of these services on maternal and infant health. Although Brazil has achieved widespread coverage of prenatal care, the quality and comprehensiveness of such services remain inconsistent across different regions and populations. The studies incorporated in this review consistently demonstrate that neonatal outcomes vary not only between those with access to prenatal care and those without but also among women who receive care of differing quality.

By examining these disparities, we uncover patterns that extend beyond mere access, highlighting how delays in initiating care, insufficient consultation frequency, and incomplete clinical or laboratory follow-up can contribute to preventable adverse outcomes. The following sections summarize these associations across key neonatal health indicators reported in the literature, reinforcing the urgent necessity to enhance both the reach and effectiveness of prenatal care in Brazil.

#### 4.11.1. Congenital Infections and Vertical Transmission Outcomes

A pervasive finding across various studies indicates a marked increase in congenital infections, particularly syphilis, correlating with inadequate prenatal care access. Comparative analyses between cohorts receiving timely screening and treatment versus those lacking such interventions highlight significant disparities in outcome severity, manifesting as elevated rates of vertical transmission, stillbirth, and neonatal hospitalizations. Crucially, the deficiency in prenatal care extends beyond mere absence; it encompasses systemic failures to incorporate infectious disease management into standard care protocols. Examples include delayed maternal testing and insufficient partner treatment, both of which intensify adverse maternal-fetal transmission outcomes. This observation underscores the necessity for not only access to prenatal care but also the implementation of comprehensive and timely infectious disease protocols to mitigate preventable pathogen transmission to the fetus.

#### 4.11.2. Prematurity and Low Birth Weight

Multiple studies analyzed in this review consistently document that inadequate prenatal follow-up correlates with higher incidences of preterm birth and low birth weight. These adverse outcomes are frequently attributed to the inability to promptly identify and manage maternal risk factors, such as hypertensive disorders, infections, and nutritional deficits, during early gestation. The limitations of prenatal care reduce critical opportunities for timely interventions, which could include pharmacological management, nutritional counseling, or strategic delivery planning. These results reinforce the concept of prenatal care functioning not merely as a monitoring mechanism, but as a proactive preventive strategy that can influence risk trajectories throughout gestation.

#### 4.11.3. Neonatal Mortality and Severe Adverse Outcomes

Data derived from the studies reviewed indicate a significant elevation in neonatal mortality rates among women who received inadequate or no prenatal care. In many instances, these fatalities stemmed from complications that are preventable through early detection and effective management, such as untreated infections, fetal growth restrictions, or delays in referral to specialized services. Furthermore, the absence of prenatal care records at delivery time frequently hinders clinical decision making during obstetric emergencies, subsequently leading to suboptimal neonatal outcomes. These associations highlight structured prenatal care as a crucial determinant of neonatal survival, particularly within at-risk populations.

#### 4.11.4. Apgar Scores, Birth Complications, and Care Completeness

In addition to severe outcomes like neonatal mortality and prematurity, studies revealed significant variations in Apgar scores and birth conditions contingent upon the adequacy of prenatal care. Newborns from pregnancies characterized by incomplete or absent prenatal care exhibited increased frequencies of low Apgar scores at five minutes, often indicative of perinatal asphyxia or inadequate obstetric intervention. These findings were frequently linked to insufficient monitoring of fetal development and a lack of preparation for high-risk deliveries. Notably, some studies suggested that even within cohorts deemed to have “sufficient” prenatal visits, low standards of care—characterized by missing examinations or inadequate patient education—were still correlated with adverse neonatal outcomes, illustrating the necessity of assessing care quality beyond mere visit frequency.

Thus, it is inferred that adequate prenatal care influences neonatal and infant outcomes cumulatively, as it not only directly impacts each clinical outcome but also because these outcomes are interrelated and mutually influence one another. Therefore, based on the findings of this review, inadequate prenatal care can be identified as a more proximal risk factor for neonatal and infant health.

To optimize the quality of prenatal care, it is essential to adopt standardized protocols for early screening and risk stratification across all tiers of healthcare delivery. Research demonstrates that late detection of maternal and fetal conditions, such as gestational syphilis, hypertensive disorders, and risks for preterm labor, is frequently attributed to the lack of structured screening frameworks. Implementing consistent assessment protocols during the first trimester—encompassing thorough medical histories, laboratory evaluations, and risk classification—can enable timely referrals and interventions, thereby reducing the incidence of avoidable complications.

A review of multiple studies, particularly those focused on congenital syphilis, highlights significant deficiencies in the integration of prenatal care with infectious disease management. Despite regular attendance at prenatal consultations, a substantial number of pregnant patients fail to receive adequate screening, accurate diagnosis, or timely treatment for infections. To address this issue, a crucial recommendation is the enhanced integration of prenatal care with infectious disease protocols. This will ensure that rapid testing, confirmatory diagnostics, and follow-up treatment are systematically embedded within prenatal workflows. Strengthening this interface is expected to decrease the rates of vertical transmission of infectious diseases substantially. Similarly, the review highlights a significant barrier in the management of sexually transmitted infections during pregnancy: the inadequate involvement of sexual partners. Research on congenital syphilis consistently demonstrates suboptimal rates of partner treatment, which exacerbates the risk of maternal reinfection and adversely affects neonatal outcomes. Therefore, public health strategies must prioritize the inclusion of partners in sexually transmitted infection-related prenatal interventions. This approach should encompass comprehensive counseling, testing, and treatment for partners, while promoting a care model that views the couple as a cohesive unit within the framework of reproductive health.

Another relevant aspect sparked by this review is that the sheer number of prenatal visits is an inadequate measure of care quality. Numerous studies have shown poor outcomes even with sufficient visit counts, underscoring the need for more comprehensive quality metrics. We suggest adopting multidimensional indicators that evaluate the completeness of prenatal care. This should include factors such as the timing of initiation, the range of laboratory tests conducted, the provision of health education, and the availability of psychosocial support. By implementing these indicators, a more accurate assessment of prenatal services can be achieved, ultimately guiding targeted improvements.

Lastly, persistent regional disparities in Brazil underscore the critical need for enhancing community health outreach, especially in rural and economically disadvantaged areas. Factors such as geographic remoteness, inadequate health literacy, and limited resource availability contribute to delays in initiating prenatal care or total non-compliance. To address these issues, it is essential to expand the roles of community health agents, family health teams, and mobile clinics. These initiatives can facilitate earlier identification of pregnancy, bolster adherence to care protocols, and effectively connect pregnant women to necessary healthcare services. Additionally, outreach programs must incorporate culturally tailored educational efforts and proactive case-finding strategies to enhance both the coverage and efficacy of prenatal services.

## 5. Limitations of the Present Report

This review has important limitations. First, by design, the evidence base was restricted to studies conducted within Brazil focus on literature published in Portuguese since 2018, which constrained the scope of data. Although this strategy was chosen to capture the specific organization, policies, and constraints of the Brazilian health system, it necessarily excludes a substantial body of international literature in other languages and may have led to the omission of relevant findings. This approach limited the breadth of perspectives and excluded international comparative evidence. Future reviews should broaden the scope by including studies from different languages and time periods, enabling more comprehensive analyses and cross-national comparisons. Second, heterogeneity in how “prenatal care adequacy” was defined across studies, limited cross-study comparability and precluded a quantitative synthesis. Third, the included studies varied widely in design, sampling frames, and sample sizes, and many relied on routine data with uneven completeness, which reduces the precision and generalizability of estimates. Fourth, as the corpus is predominantly observational, residual confounding by socioeconomic, access, and clinical factors cannot be ruled out, and causal inference remains limited. Finally, we did not apply a formal risk-of-bias tool, and we cannot exclude publication bias, especially in outcomes represented by few studies.

These constraints were particularly evident in outcomes such as congenital syphilis, neonatal near miss, infant mortality, and prematurity, where discrepant operational definitions and measurement choices complicated direct comparisons. Even so, examining distinct components of each indicator allowed a broader view of the care processes most likely to influence outcomes.

Despite these limitations, this review offers a comprehensive synthesis of Brazilian observational studies that explicitly relate prenatal care adequacy to neonatal and infant outcomes. Across regions and populations, consistent patterns emerged: delays in initiating care, insufficient consultation frequency, fragmented follow-up, incomplete diagnostic work-ups, and limited partner management for sexually transmitted infections were associated with higher risks of preterm birth, low birth weight, neonatal mortality, congenital infections, and low Apgar scores. Importantly, several studies documented that women may attend appointments yet receive incomplete content, underscoring that visit counts alone are not a reliable proxy for quality.

Taken together, the evidence supports a shift from coverage-based metrics toward standardized, multidimensional indicators of prenatal care content and continuity. Future studies should adopt harmonized adequacy frameworks, report adjustment for key confounders, apply formal quality appraisal, and, where feasible, enable meta-analytic synthesis. Addressing these methodological gaps will strengthen inference and better inform policy and practice aimed at improving maternal and infant outcomes in Brazil and in comparable middle-income settings.

## 6. Conclusions

This review shows that, in Brazil, high prenatal coverage has not been sufficient to consistently improve fetal and infant outcomes because quality of care, not visit counts alone, is the decisive lever. Across the included studies, delays in initiating care, incomplete testing, fragmented follow-up, insufficient counseling, and limited partner management for sexually transmitted infections were repeatedly associated with preterm birth, low birth weight, low five-minute Apgar scores, congenital infections (notably syphilis), neonatal near miss, and mortality. These patterns were observed across different regions and populations, indicating system-level gaps rather than isolated exceptions.

## Figures and Tables

**Figure 1 healthcare-13-02414-f001:**
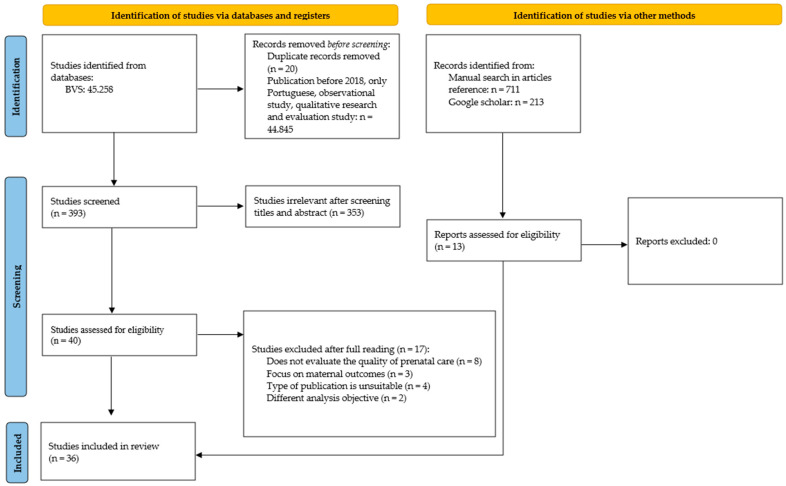
Flowchart of the study selection process. Abbreviations: BVS—Biblioteca virtual em saúde.

**Figure 2 healthcare-13-02414-f002:**
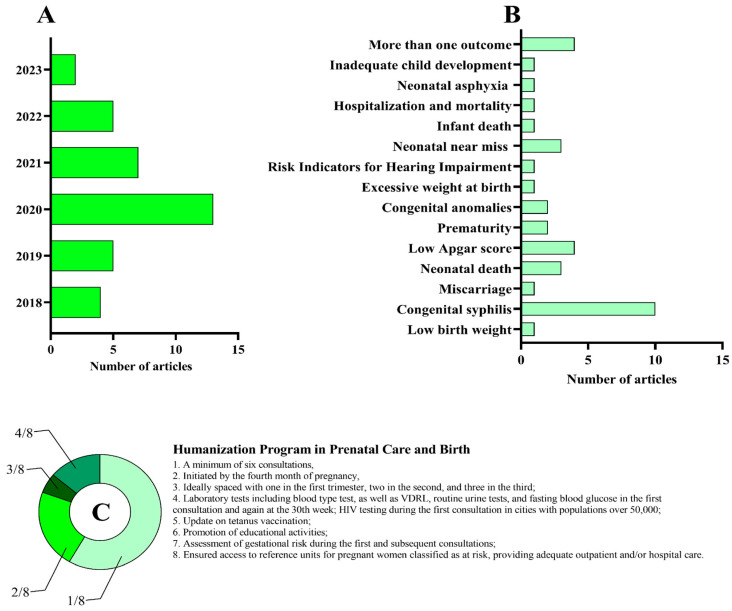
Distribution of the selected articles categorized by their year of publication (**A**), outcome (**B**), and humanization program in prenatal care and birth criteria (**C**). In panel (**C**), the articles were classified based on the number of satisfied criteria: one criterion—21/36; two criteria—8/36; three criteria—2/36; and four criteria—5/36.

**Figure 3 healthcare-13-02414-f003:**
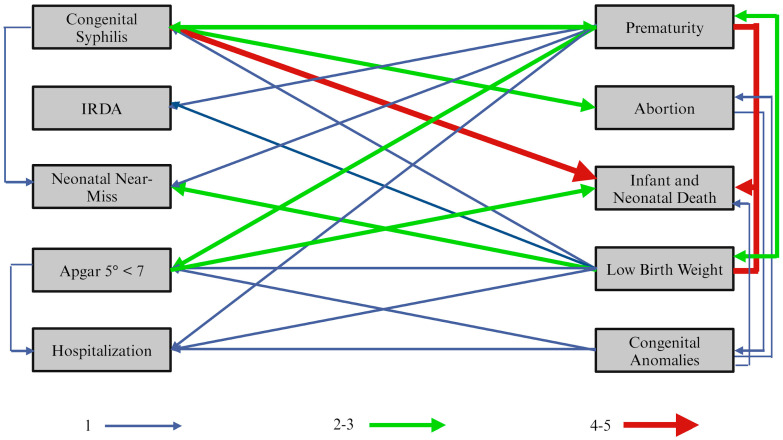
The diagram illustrates the intricate cause-and-effect relationships that influence infant and neonatal outcomes, with arrow thickness indicating the volume of studies linking these outcomes. The width of the arrows and lines is directly proportional to the quantity of correlated articles. Congenital syphilis, prematurity, and low birth weight are emphasized as critical contributors to both infant and neonatal mortality, with five studies collectively examining these interrelations. Notably, congenital syphilis is linked with prematurity, while low birth weight is associated with prematurity in two separate studies. Additionally, congenital syphilis is identified as a causal factor for abortion in three pieces of literature, and prematurity is associated with low Apgar scores in two studies. Two studies identified low birth weight as a contributor to neonatal near-miss incidents. The relationship between low Apgar scores and subsequent neonatal and infant mortality is underscored in two publications. Other associations among outcomes are reported in individual studies.

**Table 1 healthcare-13-02414-t001:** Descriptive statistics of the selected studies.

Author (Year)	Study Design	Sample Size	Prenatal Care Quality Metric	Newborn’s Outcome
Almeida et al. (2020) [14]	Cross-sectional study	326 children	Number of prenatal visits < 6 or >; or =6.	Excessive weight gain among preschoolers
Araújo et al. (2021) [15]	Cross-sectional study	478 babies with congenital syphilis	Attended prenatal care, number of visits, syphilis testing, prenatal treatment of the pregnant woman, and partner’s treatment	Prematurity of babies with congenital syphilis
Assis et al. (2022) [16]	Cross-sectional study	4571 puerperal women and newborns	Prenatal care starting before 12 weeks of pregnancy, the number of visits, and routine exams	Neonatal near miss
Belfort et al. (2018) [17]	Cross-sectional study	751 teenagers	Number of visits, gestational age at the first visit	Low birth weight
Brito et al. (2022) [18]	Cross-sectional study	1219 women and children	Start of prenatal care, number of prenatal visits	Low Apgar score
Caldeira et al. (2021) [19]	Epidemiological and cross-sectional study	198 pregnant women	Prenatal care location and risk, prenatal visits, and syphilis diagnosis during prenatal care	Congenital syphilis
Carvalho et al. (2020) [20]	Descriptive, documental, retrospective, and cross-sectional study	147 women with a diagnosis of miscarriage	Number of prenatal visits	Miscarriage
Favero et al. (2019) [21]	Observational and cross-sectional study	Gestational syphilis (120) and Congenital syphilis (103)	Prenatal care, timing of maternal diagnosis, and treatment initiated	Congenital syphilis
França et al. (2021) [22]	Cross-sectional study	2012, n = 304 2016, n = 243	Number of prenatal visits	Neonatal near miss
Kierenco et al. (2022) [23]	Descriptive, retrospective study with a quantitative approach	190,034 cases of congenital syphilis (2011–2020)	Women: Prenatal care, diagnosis during prenatal care, and treatment Children: age at the time of diagnosis, classification of the final diagnosis	Congenital syphilis
Magalhães et al. (2023) [24]	Cross-sectional study	1999, n = 2,808,341 DNVs, 58,961 patients presenting Apgar < 7 2018–2019: n = 5,680,092. DNVs, 52,731 presenting Apgar < 7	Number of prenatal visits	Low Apgar score
Maia et al. (2020) [9]	Case–control study	7470 cases e 24,285 controls	Number of prenatal visits	Infant death
Nascimento et al. (2020) [25]	Longitudinal cohort study	87 preterm and term babies	Number of prenatal visits (WHO): 8 visits	Risk Indicators for Hearing Impairment (RIHI)
Oliveira et al. (2019) [26]	Observational case–control study	296 cases e 329 controls	Start, number of visits, conducting exams, basic procedures, and guidance	Prematurity
Pavaneli (2022) [27]	Epidemiological retrospective cross-sectional study	118 babies	Number of prenatal visits	Low Apgar score in the fifth minute
Reis (2018) [28]	Analytical ecological study	6274 cases of congenital syphilis	Prenatal care coverage did or did not receive prenatal care	Congenital syphilis
Saloio et al. (2020) [29]	Retrospective cohort study	21,346 live births	Number of prenatal visits	Neonatal death
Silva et al. (2020) [30]	Descriptive epidemiological study	1029 cases	Start of prenatal care, inadequate treatment of syphilis	Congenital syphilis
Silva et al. (2021) [31]	Cross-sectional study with a quantitative approach	145 medical records of pregnant women with infection	Number of prenatal visits	Different fetal/neonatal outcomes
Sleujtes et al. (2018) [32]	Case–control study	162 deaths Neonates born in the same year (controls)	Number of prenatal visits	Neonatal death
Trevilato et al. (2022) [33]	Case–control study	5250 live births with congenital anomaly; 21,000 congenital anomaly	Number of prenatal visits	Congenital anomalies
Vanin et al. (2020) [34]	Case–control study	423 patients 141 cases and 282 controls	Number of prenatal visits; start of prenatal care	Late preterm
Vidal et al. (2023) [35]	Cross-sectional study	440 puerperal women	Early start of prenatal care, number of prenatal visits, immunization during pregnancy, HIV and syphilis testing	Spontaneous preterm birth, low birth weight, Apgar at 1 and 5 min
Bicalho et al. (2021) [36]	Observational, descriptive study with a quantitative approach	403 cases	Prenatal care, Timing of maternal syphilis diagnosis	Congenital syphilis
Branco et al. (2020) [37]	Retrospective epidemiological study, descriptive in nature, with a quantitative approach	603 cases	Prenatal care, Timing of maternal syphilis diagnosis	Congenital syphilis
Fernandes et al. (2020) [38]	Descriptive, quantitative, retrospective, cross-sectional study	5358 births with 108 cases of asphyxia	Number of prenatal visits	Neonatal asphyxia (Apgar score < 6 at the fifth minute)
Garcia et al. (2019) [39]	Cohort study	15,879 live births and 86 deaths	Number of prenatal visits	Neonatal death
Leal et al. (2020) [40]	Analytical observational cross-sectional study	23,894 puerperals and their liveborn or stillborn babies with birth weight ≥ 500 g and/or gestational age ≥ 22 weeks	Number of visits, type of prenatal care unit, trimester of start of prenatal care, exams, and guidance	Spontaneous preterm birth, low birth weight, intrauterine growth restriction, Apgar at fifth minute < 8, neonatal near miss, maternal near miss
Luz et al. (2019) [41]	Descriptive time series study	1,386,803 live births, 12,818 with congenital malformations	Prenatal visits	Congenital anomalies
Macêdo et al. (2020) [42]	Descriptive study	1206 women	Number of visits, location of prenatal care, and definition of appropriate treatment	Congenital syphilis
Moura et al. (2020) [43]	Cohort study	56,341 live births	Number of prenatal visits	Hospitalization and mortality
Padovani et al. (2018) [44]	Retrospective cross-sectional study	Congenital syphilis (306 cases)	Prenatal care and visits, syphilis diagnosis, and trimester of diagnosis	Congenital syphilis
Pereira et al. (2020) [45]	Prospective cohort study	48,037 newborn	Prenatal visits, record in the prenatal card at least one result from each routine exam	Neonatal near miss
Santos et al. (2019) [46]	Retrospective analytical cross-sectional study	9585 deliveries	Number of prenatal visits	Low Apgar score
Soares et al. (2021) [47]	Ecological and longitudinal study	15,050 cases of gestational syphilis and 7812 cases of congenital syphilis	Number of prenatal visits; conducting the rapid test	Congenital syphilis
Sólis-Cordeiro et al. (2021) [48]	Cross-sectional study	125 days (mother + child)	Number of prenatal visits	Inadequate child development

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
