# Peer review of "The Role of Prenatal Care in Fetal and Infant Development in Brazil: A Narrative Review"

_healthcare, 2025, doi:10.3390/healthcare13192414_

Round 1

Reviewer 1 Report

Comments and Suggestions for Authors

This is an interesting study. However, some points are needed to improve.

Author Response

# REVIEWER 1

This is an interesting study. However, some points are needed to improve.

Answer: We appreciate the Reviewer’s positive evaluation of our manuscript. Your observations contributed significantly to improving the scientific clarity, precision, and robustness of the final version of our work. All suggestions were carefully considered and addressed either directly in the revised manuscript or in the specific responses below. The changes made to the text are clearly highlighted in green in the revised version. They are annotated with indications of the respective page and line numbers for transparency and ease of verification.

Once again, we thank you for your valuable contribution to refining our study.

 I have some comments:

  1. Complete this paragraph minimum three sentences in one paragraph (Line 47-50).

Answer: Thank you for your insightful observation. We have expanded the paragraph to provide additional clarification on the multifaceted nature of prenatal care and the lack of consensus in defining adequacy. The modifications are highlighted in green (Page 2, line 43 to 51).

  1. clarify that this appointment is during pregnancy or what else? (Line 54-59)

Answer: Thank you for pointing out this imprecision. We agree that it is essential to specify that the recommended consultations refer to prenatal appointments conducted during pregnancy. The modifications are highlighted in green (Page 2, line 55 to 61).

  1. explain what the minimum standard is? (Line 62)

Answer: We thank the Reviewer for this important observation. We agree that it is necessary to clarify what is meant by “minimum standards” in the context of the Prenatal and Birth Humanization Program (PHPN). We have revised the text to specify the main elements defined by the PHPN, which include timely initiation of care, a minimum number of consultations, and completion of recommended laboratory and clinical procedures.We included a brief mention about this topic in the revised document, which is highlighted in green (page 2, line 75 to 80).

  1. explain the barriers (Line 63)

Answer: We appreciate the Reviewer for addressing this important observation. In this context, the term "barriers" is intended to refer to systemic limitations and challenges that undermine the adequacy and continuity of prenatal care. To enhance clarity, we have revised the text to outline these barriers more explicitly, including late initiation of prenatal care, insufficient completion of fundamental examinations, fragmented follow-up throughout the pregnancy and postpartum continuum, and ongoing regional and socioeconomic disparities that obstruct equitable access to services. All modifications are highlighted in green (Page 2, line 80 to 84).

  1. Complete this paragraph minimum three sentences in one paragraph. (Line 68-72)

Answer: We thank the Reviewer for this constructive suggestion. We have expanded the paragraph to better highlight the paradox of high prenatal care coverage coexisting with unfavorable outcomes and to emphasize the implications for research and policy. All modifications are highlighted in green (Page 3, line 87 to 96).

  1. add when this review was conducted? (Line 99)

Answer: Thank you for the comment. The search was conducted in July 2024 and updated in February of this year (2025). We highlighted this information in the revised manuscript (Page 4, line 133 to 134).

  1. apply format in repeat header row (table 1).

Answer: We thank the Reviewer for this technical observation. We have reformatted Table 1 to ensure that the header row is repeated across all pages where the table is displayed, in accordance with journal formatting standards.

  1. this figure should be presented earlier in the text (Figure 1)

Answer: Response: We thank the Reviewer for this helpful suggestion. We have repositioned Figure 1 so that it appears immediately after the description of the study selection process, ensuring better alignment between the text and the corresponding figure (Page 5).

  1. need to increase the font size from the figure (Figure 2)

Answer: We thank the Reviewer for this valuable observation. Following the suggestion, we have increased the font size in Figure 2 and also adjusted the font size across all figures in the manuscript to ensure readability and consistency.

  1. Complete this paragraph minimum three sentences in one paragraph. (Line 224-226)

Answer: We thank the Reviewer for this suggestion. We have expanded the paragraph to better contextualize the transition to the discussion section, emphasizing the interrelations among outcomes and the implications for healthcare quality and policy. All modifications are highlighted in green (Page 11, line 248 to 255).

  1. moved into the end of sentence (Line 239)

Answer: Thank you for the comment. We moved the citation to the end of the sentence.

  1. please revise the citation format as we did not present the year of study (Line 270-271)

Answer: We thank the Reviewer for pointing out this formatting issue. We have revised the citation style to align with the journal’s guidelines, removing the year of publication and presenting only the reference numbers.

  1. we did not think that it needs to present this figure in the study (Figure 3 – Line 618)

Answer: We appreciate the Reviewer’s observation. However, we respectfully maintain that Figure 3 is a valuable addition to the manuscript, as it offers a dynamic and visual representation of the interrelations among the outcomes identified in the included studies. This figure enhances the reader’s understanding by illustrating how adverse outcomes cluster and interact rather than occurring in isolation, thereby reinforcing one of the central themes of our review. Additionally, we would like to point out that the manuscript features relatively few figures overall, and those included were carefully selected as essential components to facilitate a more cohesive and meaningful reading experience, particularly in the context of a review article. For these reasons, we have decided to retain Figure 3 in the revised version.

Reviewer 2 Report

Comments and Suggestions for Authors

Thank you for the opportunity to review this study entitled “The Role of Prenatal Care in Fetal and Infant Development in Brazil: A Narrative Review” (healthcare-3868406).
The paper presents a review focusing on how the adequacy of prenatal care relates to fetal and child health. Quality gaps in service delivery are also identified.

In my opinion, the research topic is relevant and interesting. At the same time, some issues need to be addressed before the paper is suitable for publication.

  • Abstract: This section should briefly summarize the main points in order to capture the reader’s attention and encourage them to continue reading the paper. In its current form, it is too long and risks having the opposite effect.
  • Keywords: Please arrange the keywords in alphabetical order.
  • Introduction: This section should be significantly expanded by introducing more scientific evidence that highlights the importance of this focus, and by clarifying why a review that synthesizes these findings is needed. At present, most of the space is devoted to the Brazilian context. While this is relevant, space should also be given to the broader background.
  • Methods: The authors identify this paper as a narrative review. However, given the current state of international research, this type of review has important gaps that are at least partially addressed by PRISMA guidelines. Some elements already recall PRISMA standards, therefore I suggest that the authors consider transforming this into a PRISMA review.
  • Methods: In addition, the inclusion criteria appear overly restrictive (only Portuguese-language articles and only from 2018 onwards). The choice should be better justified, as it excludes a large body of international comparative literature that could enrich the perspective.
  • Page 10, lines 224–226: This part seems unnecessary and redundant. I suggest removing it.
  • Discussion: From page 19 onwards, a new list is introduced and numbered again. Since this is confusing, I recommend using a different listing system.

Best wishes

Author Response

# REVIEWER 2

Thank you for the opportunity to review this study entitled “The Role of Prenatal Care in Fetal and Infant Development in Brazil: A Narrative Review” (healthcare-3868406).
The paper presents a review focusing on how the adequacy of prenatal care relates to fetal and child health. Quality gaps in service delivery are also identified.

In my opinion, the research topic is relevant and interesting. At the same time, some issues need to be addressed before the paper is suitable for publication.

Answer: We appreciate the Reviewer’s positive evaluation of our manuscript. Your observations contributed significantly to improving the scientific clarity, precision, and robustness of the final version of our work. All suggestions were carefully considered and addressed either directly in the revised manuscript or in the specific responses below. The changes made to the text are clearly highlighted in green in the revised version. They are annotated with indications of the respective page and line numbers for transparency and ease of verification.

Abstract: This section should briefly summarize the main points in order to capture the reader’s attention and encourage them to continue reading the paper. In its current form, it is too long and risks having the opposite effect.

Answer: We appreciate the Reviewer's insightful comment. We concur that the abstract should be both concise and engaging to effectively highlight the most pertinent aspects of the study. Therefore, we have revised the abstract to shorten it and enhance its focus, while preserving a clear summary of the background, objectives, methods, key findings, and conclusions. The revised version underscores the paradox of high prenatal care coverage alongside ongoing adverse outcomes, highlights the main findings related to the inadequacy of care, and discusses the implications for maternal and child health policies. All modifications are highlighted in green (Page 1).

Keywords: Please arrange the keywords in alphabetical order.

Answer: Thank you for the comment. In the revised manuscript, keywords were arranged in alphabetical order. All modifications are highlighted in green (Page 1).

Introduction: This section should be significantly expanded by introducing more scientific evidence that highlights the importance of this focus, and by clarifying why a review that synthesizes these findings is needed. At present, most of the space is devoted to the Brazilian context. While this is relevant, space should also be given to the broader background.

Answer: We thank the Reviewer for this important suggestion. We agree that the Introduction should provide a broader scientific background, highlighting international evidence and clarifying the rationale for synthesizing Brazilian findings. To address this, we have included new sentences that emphasize the importance of systematic reviews and narrative syntheses in contextualizing these outcomes, and we explain why a focus on the Brazilian context remains relevant despite high national coverage. All modifications are highlighted in green (Page 2, line 62 to 74, and Page 3, line 114 to 117).

Methods: The authors identify this paper as a narrative review. However, given the current state of international research, this type of review has important gaps that are at least partially addressed by PRISMA guidelines. Some elements already recall PRISMA standards, therefore I suggest that the authors consider transforming this into a PRISMA review.

Answer: We thank the Reviewer for this thoughtful suggestion. We agree that systematic reviews following PRISMA guidelines provide a higher level of methodological rigor and transparency. However, our work was intentionally designed as a narrative review, which allowed us to map and synthesize the evidence while providing a contextualized discussion of the Brazilian scenario. Although we incorporated certain methodological steps inspired by PRISMA (e.g., structured database search, independent screening, and data extraction), the scope and objectives of the study do not fully meet the requirements of a systematic review. We recognize the value of conducting a systematic review on this topic and consider this narrative review an important preliminary step to inform such an effort. In future work, we plan to conduct a systematic review specifically focused on prenatal care adequacy in Brazil, expanding the scope to include all available studies without restrictions on language or time period, thereby enabling a comprehensive timeline of evidence.

Methods: In addition, the inclusion criteria appear overly restrictive (only Portuguese-language articles and only from 2018 onwards). The choice should be better justified, as it excludes a large body of international comparative literature that could enrich the perspective.

Answer: We thank the Reviewer for this important observation. We agree that the inclusion criteria applied in this review, which restricted the scope to Portuguese-language articles published from 2018 onwards, constitute a limitation. This decision was a strategic choice intended to emphasize evidence directly reflecting the Brazilian healthcare context and to capture the most recent literature aligned with national programs and policies. By doing so, we aimed to highlight local challenges that are sometimes underrepresented in international comparative reviews.

We acknowledge that this approach constrained the breadth of our data and may have excluded valuable international perspectives. For this reason, we have explicitly recognized it as a limitation in the manuscript. Furthermore, we plan to conduct a future systematic review that will broaden the scope by including studies published in all languages and without temporal restriction, thereby enabling a comprehensive timeline and cross-national comparisons. All modifications are highlighted in green (Page 23, line 764 to 766, and line 769 to 772).

Page 10, lines 224–226: This part seems unnecessary and redundant. I suggest removing it.

Answer: We appreciate the Reviewer’s suggestion and acknowledge the concern regarding redundancy. However, we respectfully maintain that retaining this paragraph is crucial for the manuscript's overall structure. Another reviewer had previously suggested expanding this section to create a clearer transition into the Discussion, and we believe that the current version effectively addresses that need. Specifically, this introductory paragraph serves as a guidepost for readers, elucidating how the findings are integrated within the Brazilian healthcare context and the organization of the subsequent discussion. We therefore believe that its maintenance enhances readability and aids in the interpretation of the review’s results.

Discussion: From page 19 onwards, a new list is introduced and numbered again. Since this is confusing, I recommend using a different listing system.

Answer: We thank the Reviewer for this helpful observation. We agree that clarity and consistency in numbering are essential. To address this, we reformatted the section by presenting the items as subheadings under “4.11. Expert opinion and perspectives”, numbered sequentially as 4.11.1, 4.11.2, 4.11.3, and so forth. This approach reiterates that these items are subcomponents of the same topic, ensuring coherence with the structure adopted in the other sections of the manuscript. All modifications are highlighted in green (Pages 21 and 22).

Round 2

Reviewer 1 Report

Comments and Suggestions for Authors

We accept this current form of revision